# Towards Antibiotic Synthesis in Continuous-Flow Processes

**DOI:** 10.3390/molecules28031421

**Published:** 2023-02-02

**Authors:** Marziale Comito, Riccardo Monguzzi, Silvia Tagliapietra, Giovanni Palmisano, Giancarlo Cravotto

**Affiliations:** 1Dipartimento di Scienza e Tecnologia del Farmaco, University of Turin, Via Pietro Giuria 9, 10125 Turin, Italy; 2Research and Development, ACS Dobfar SpA, Via Paullo 9, 20067 Tribiano, Italy; 3Dipartimento di Scienza e Alta Tecnologia, University of Insubria, Via Valleggio 9, 22100 Como, Italy

**Keywords:** flow chemistry, antibiotics, continuous process, drug synthesis, process control, industrial application, miniaturization

## Abstract

Continuous-flow chemistry has become a mainstream process and a notable trend among emerging technologies for drug synthesis. It is routinely used in academic and industrial laboratories to generate a wide variety of molecules and building blocks. The advantages it provides, in terms of safety, speed, cost efficiency and small-equipment footprint compared to analog batch processes, have been known for some time. What has become even more important in recent years is its compliance with the quality objectives that are required by drug-development protocols that integrate inline analysis and purification tools. There can be no doubt that worldwide government agencies have strongly encouraged the study and implementation of this innovative, sustainable and environmentally friendly technology. In this brief review, we list and evaluate the development and applications of continuous-flow processes for antibiotic synthesis. This work spans the period of 2012–2022 and highlights the main cases in which either active ingredients or their intermediates were produced under continuous flow. We hope that this manuscript will provide an overview of the field and a starting point for a deeper understanding of the impact of flow chemistry on the broad panorama of antibiotic synthesis.

## 1. Introduction

Advances in industrial organic synthesis are essential for the successful commercialization of innovative and efficient chemical–pharmaceutical manufacturing. Achievements, from the discovery of salvarsan to advanced therapy medicinal products (ATMPs), would have been impossible without cutting-edge technology and interdisciplinary collaboration [1,2,3,4]. The new technologies and modern trends in the synthesis of drugs and natural products that have been developed by academia and industry are opening up opportunities on a scale previously considered unattainable in most laboratories and production lines. The use of high-throughput and breakthrough technology platforms, particularly flow chemistry and process analytics (PAT), is representative of the endless potential in the pharmaceutical field and the improvements over the current state that are possible (Figure 1) [4,5,6,7,8,9,10,11,12,13,14,15,16,17,18,19,20,21,22,23,24,25].

In an era where sustainability is driving industrialization and innovation, in accordance with environmental friendliness and green chemistry concepts [26,27,28,29,30,31,32,33,34,35,36,37], the pharmaceutical industry is at the forefront of embracing and leading change. The pharmaceutical industry’s mission is to provide patients with new medicines to help them live longer and healthier lives by creating small molecules in accordance with drug-development protocols. Until not so long ago, drug companies ignored risks to workers and the environment. Today, their approach has changed completely. In 2020, small-molecule drugs accounted for approximately USD 478 billion in sales in the global pharmaceutical markets, and this figure is expected to grow at 7% annually through to 2024 [38].

In 2005, the American Chemical Society (ACS) Green Chemistry Institute (GCI) and the most important global pharmaceutical corporations set up the ACS GCI Pharmaceutical Roundtable. Their aim was to encourage the integration of green chemistry and green engineering into the synthesis of small molecules [39,40,41]. This concept has influenced all phases of drug development over the last twenty years, from preclinical to commercial stages, and has become a successful feature for new molecular entities (NMEs). At the same time, the opportunities presented by renewing old synthetic routes with new technologies have grown into a vast research field.

Of the many emerging technologies available, continuous manufacturing, which is also known as continuous processing or continuous-flow chemistry, has become the mainstream in the synthesis of active pharmaceutical ingredients (APIs). Its impact on the life cycles of drugs has been so overwhelming that the Food and Drug Administration (FDA) and European Medicines Agency (EMA) have recently drawn up guidelines for this manufacturing type [42,43]. Although publications on flow chemistry have exponentially grown in number over the last two decades, including assessments of pros and cons [11,44,45,46], and despite its use being quite commonplace in many industries, the pharmaceutical world is recalcitrant to adopt it, and batch manufacturing remains king. The availability of standard reactors together with the simplicity, versatility and flexibility of their use means that old habits die hard. Although the positive impact of flow mode is now recognized, its application on an industrial scale is still seen as being the game changer that is too volatile to welcome. The industry’s hesitance to embrace continuous-flow processes is understandable precisely because the majority of publications derive from within the academic sphere, many processes are relatively untested and regulatory guidance is too young. In order to extend the scope of these technologies, companies must be sure of their suitability for specific business needs, including an awareness of their operational advantages, as well as the challenges they pose [47,48,49,50,51].

Chemical reactions in discontinuous processes occur in large vessels for a given time before the product is crystallized, discharged, analyzed and, eventually, purified. If a problem emerges during synthesis, or if the product does not comply with standard quality guidelines, production is compromised, causing undesirable losses in money and time. Continuous manufacturing runs constantly until a project is complete, slashing manufacturing times and avoiding breaks between the steps. Given that reactions take place on a much smaller scale, only small amounts of product are lost if the process fails. The automated nature of continuous processes minimizes fluctuations in reaction conditions (e.g., temperature, pressure and reaction time) and reduces human error compared to batch manufacturing, saving assets. For the same reasons, chemists can easily control reactions in continuous flow (also combining photo- and electrochemistry), whereas this is a critical issue in batch mode because of the extreme conditions and the presence of highly reactive and unstable intermediates [52,53,54,55,56,57,58,59,60,61,62,63,64,65,66]. Miniaturization intrinsically improves synthesis due to the excellent mass and heat transfer that it provides, also meaning that less laboratory and industrial space is required. Modularization allows integrated synchronized operations to be performed, facilitating adaptability to different pharmaceutical processes. The closed architecture of these systems provides safer working conditions as it eliminates direct contact with hazardous chemicals and avoids production-chain incidents. Integration with process analytical technologies (PAT) and purification modules has boosted this technology’s status, making the drug-production process telescopic, increasing production capability while retaining substance quality. Green chemistry concepts are met because the product does not need to be isolated and stored before use in a subsequent step, as it can directly flow into the subsequent reactor for another synthesis or into another module for another operation [67,68,69,70,71,72,73,74].

There are many reasons for favoring and adopting continuous manufacturing in the pharmaceutical world (Figure 2), including reasons that support heavy investment in the production of drugs and precursors. In this short review, we present the flow-mode applications of the synthesis of antibiotics and their building blocks. We have covered the period of 2012–2022, as we highlight the key points and merits of applying this new technology for these important drugs. Most of the small molecules studied are off patent, are characterized by chemistry that was developed many years ago and have only been relaunched in some cases [75,76,77,78,79,80,81]. We hope that flow chemistry can revive them with renewed vigor.

## 2. Discussion

Table 1 lists all of the antibiotics for whose synthesis flow chemistry has been applied. They will be discussed in the subsequent sections.

## 3. Antibiotic Synthesis in Flow Mode

### 3.1. Cefotaxime

Cefotaxime (**1**) is a β-lactam antibiotic classified as a third-generation cephalosporin, was first synthesized in 1976 and was commercialized by 1980 under the brand name Claforan^TM^. It was approved by the FDA to treat Gram-positive, Gram-negative and anaerobic bacteria.

Its broad-spectrum activity is useful in treating complicated urinary-tract infections, lower-respiratory-tract infections, bacteremia, meningitis, uncomplicated gonorrhea, skin and soft-tissue infections, and obstetric and gynecological infections. Its activity takes place via linkage to the penicillin-binding proteins (PBPs) via its β-lactam ring and by inhibiting the transpeptidation step in the peptidoglycan cell wall. It appears on the World Health Organization’s List of Essential Medicines and is available for intramuscular and intravenous administration. It is distributed in powder form in 500 mg, 1 g, 2 g and 10 g vials or in a premixed solution for injection of 1 g and 2 g [92,93,94,95,96,97,98].

The processes to produce cephalosporins, including cefotaxime (**1**), were developed over thirty years ago and persist to this day. In the current context, which is characterized by strict concerns over worker safety and respect for environmental and energy savings, flow chemistry represents an attractive option for the synthesis of these drugs [99,100].

Pieper et al. have published an interesting study on the synthesis of this β-lactam antibiotic in flow mode and have made comparisons to batch mode [82]. The synthesis involved the amidation step between 7-aminocephalosporanic acid (7-ACA) and (Z)-(2-aminothiazol-4-yl)-methoxyimino acetic acid under activation by 4-toluenesulfonyl chloride, as represented in Figure 1.

In this system, 7-ACA was dissolved and stored in dichloromethane with triethylamine at 0 °C in a vessel. In another vessel, the mixed anhydride suspension was formed in dimethylacetamide (DMAc) at −11 °C. This vessel was constantly stirred to avoid sedimentation and was stored for a maximum of 1 h to respect the stable hold time. Peristaltic pumps (P100 and P200) transported the raw-material solution to the Y-shaped polypropylene mixer (M100) in flow towards the reactor. The flow reactor was a fluorinated ethylene propylene (FEP) tube of 10 mL with an inner diameter of 4 mm that was submerged in butyl glycol, used as a cooling material. The cooling batch was tempered with a cooling jacket through which butyl glycol was constantly pumped. The flow rate was set up at 5 mL/min to prevent sedimentation effects that may arise from the insoluble mixed anhydride. The output was collected in a vessel so that the product could be analyzed in solution.

With this methodology, cefotaxime (**1**) was generated at a yield of 80.9% to 7-ACA, working at −10 °C and using a residence time of 1 min. Higher reaction temperature (+20 °C) led to higher 7-ACA conversion, but lower product yield because of the degradation of the cephalosporin nucleus. Higher residence times decreased 7-ACA conversion and, consequently, cefotaxime (**1**) yield. This methodology allowed much shorter reaction times (1 min) to be used, compared to the 30 min needed in the batch synthesis. Furthermore, a more convenient temperature (−10 °C), than the −30 °C used for batch mode, is possible, providing a further advantage in energy savings and costs. The space-time yield was nearly 400 times higher than when the reaction is performed in a reactor vessel and efficient heat distribution corroborated the technology’s safety.

This result clearly shows the attractive features that the flow process possesses compared to the analogous batch route, although the yield was slightly lower. In order to boost productivity and overcome this shortcoming, the authors have proposed operating several identical systems in parallel [101], which would consolidate the great capabilities of flow chemistry.

### 3.2. Cephalexin

Cephalexin (**2**) is one of the most widely prescribed β-lactam antibiotics in the United States of America, with more than 7 million prescriptions being made in 2020 [102,103]. It is a first-generation cephalosporin discovered in 1967 and marketed under the brand names Keflex^TM^ and Ceporex^TM^ since 1969. It is used against Gram-positive and some Gram-negative bacteria, particularly *E. coli*, *Proteus mirabilis* and *Klebsiella pneumoniae*. It is administered orally as either 250 mg or 500 mg capsules to treat urinary-tract and respiratory infections. It appears on the World Health Organization’s List of Essential Medicines [98,104,105,106,107,108,109].

Vobeckà et al. have reported a continuous-flow process for the production of cephalexin (**6**), with excellent outcomes [83]. The product was synthesized from phenylglycine methyl ester (PGME) and 7-aminodesacetoxycephalosporanic acid (7-ADCA) using penicillin acylase (PA) as the enzyme in a kinetic regime (Figure 3). This regime was necessary to achieve a high cephalexin (**2**) yield in a water medium, as this is difficult to achieve in the thermodynamic process.

As depicted in Figure 2, the authors employed apparatus that is characterized by an aqueous two-phase system (ATPS) that forms two phases in the flow reactor, which acts as a reaction-separation environment. This system, used in a microfluidic arrangement, guaranteed the in situ extraction of cephalexin (**2**), as well as facilitating enzyme recycling, the addition of fresh reactants and the presence of a uniform reaction mixture. The ATPS consisted of 15 wt% of polyethylene glycol (PEG), with molecular weight ranging from 2000 g mol^−1^ to 4000 g mol^−1^, 12 wt% of phosphates, to ensure pH = 7.0, and 73 wt% of water. This composition granted cephalexin (**2**) high affinity to the top phase, which split from the PA-containing bottom phase. The reaction mixture contained PGME and 7-ADCA, in a molar ratio of 3:1, which dissolved in ATPS at concentrations of 150 mM and 50 mM, respectively. The free enzyme in water had a concentration of 10 μL for 1 mL of reaction mixture and an activity of 2.88 kU mL^−1^. The reaction temperature was set to 30 °C by immersing the flow reactor in a water bath.

Specifically, the reactants were dosed from the vessel (d) into the microcapillary reactor, (b), which has an inner diameter of 0.8 mm and a length of 87 cm, via a three-way PEEK connector (a). The PA was pumped from the recycle-system vessel (e) into the reactor. The flow rate was set to the value that provided a residence time of 20 min. The phases were separated in a gravity settler (c) that was placed at the reactor outlet and was made of a 10 mL plastic syringe, without a vertically positioned piston and with the needle oriented downwards. Filter paper was placed at the bottom of this settler for the continuous removal of any precipitated phenylglycine (PG). The top phase was withdrawn from the top part of the settler (c) by a peristaltic pump (g) and collected in a product vessel (f). The bottom phase was pumped from the settler (c) through the filter paper into the reservoir (e), with continuously stirred enzyme recycling. To avoid PG clogging in the reaction mixture, the authors added a microdialyzer (h), operating in counter-current flow, that was closed from both sides using ad hoc made Plexiglas ports (k). In this way, the enzyme solution was restored fresh to the reservoir (e). The dialysate solution was collected in a waste reservoir (i), while a fresh phase was pumped from the reservoir (j) into the intertubing of the microdialyzer to clean the enzyme solution.

Using this integrated microfluidic platform, the authors obtained a cephalexin (**2**) yield of 80%, with respect to 7-ADCA, whereas the yield in batch mode was 75% under the same conditions. In addition, this technology was able to operate continuously, generating new C-N bonds, for at least 5 h. The optimization of reaction conditions, performed first in batch mode and then in flow mode, was fundamental to the success of this approach. Enzyme recycling, considerable time and cost reductions and high productivity are the advantages of this protocol.

### 3.3. Ciprofloxacin Hydrochloride

Ciprofloxacin hydrochloride (**3**) is an antibiotic agent in the fluoroquinolone class that is used to treat bacterial infections caused by Gram-negative bacteria, such as urinary-tract infections and pneumonia. It is also used against sexually transmitted infections (gonorrhea and chancroid) and lower-respiratory-tract infections. It was patented in 1983 by Bayer and approved in 1987 by the FDA. It was marketed with the brand name Cipro^TM^ and reached a sales peak of USD 2 billion in 2001. It appears on the World Health Organization’s List of Essential Medicines [98,110,111,112,113,114,115,116].

First, Lin and coworkers and, secondly, Armstrong et al. demonstrated, in two different manuscripts, an effective strategy for the total synthesis of ciprofloxacin hydrochloride (**3**) in flow mode [84,85]. In 2017, Lin et al. [84] applied flow-chemistry concepts to all six chemical reactions as they developed a telescopic process in five flow reactors (Figure 3). They obtained ciprofloxacin hydrochloride (**3**) in a 60% overall yield, whereas the yield was 49% in the Bayer batch process. This result was achieved in a total residence time of only 9 min, whereas more than 24 h is required in batch mode, making this one of the longest linear sequences without the flow being interrupted in any workup.

Starting from 2,4,5-trifluorobenzoyl chloride and ethyl-3-(dimethylamino)acrylate, they optimized all of the reaction parameters to perform the process while reducing time and waste. Notably, inline acylation with acetyl chloride in the third flow reactor allowed the dimethylamine byproduct to be converted into dimethylacetamide (DMAc), thus overcoming the low-yield problem in the fourth step [117]. To avoid intermediate clogging at the inlet stream of the fifth flow reactor, the authors warmed the solution, due to its low solubility, to continue the synthesis.

In 2021, inspired by Lin’s work and by Massachusetts Institute of Technology (MIT) researchers [118,119,120,121], Armstrong and coworkers developed a ciprofloxacin flow synthesis with an industrial perspective (Figure 4). They scaled-up the process 1.5-to-2-fold to obtain enough crude product for 1000 tablets in 24 h. They demonstrated high reproducibility and robustness, providing concrete improvements to previous works, as they continuously ran the first three steps for 22 h and the last two steps for 10 h. This work represents the perfect combination of design of experiments (DOE), PAT and flow chemistry.

The authors determined and optimized operating and stoichiometric parameters with DOE in order to minimize impurities and increase step yields. They implemented continuous liquid–liquid extraction (CLLE, Figure 4) to remove dimethylamine without using acetyl chloride as a catching agent, meaning that the number of reactions could be reduced from five to four. This operation increased the concentration of the second intermediate in the organic stream and enabled the use of 2-methyltetrahydrofuran as a green solvent. The selected solvent system, with 1 M hydrochloric acid, was operated at a 5:2:3 ratio for the aqueous solution–organic/solution–reaction stream. 1,8-Diazabicyclo [5.4.0]undec-7-ene (DBU) was replaced with tetrabutylammonium hydroxide (TBAOH), meaning that blockages in the lines and the formation of impurities were avoided. The finalized process conditions afforded a yield of 91 ± 2% for the first two steps, providing the second intermediate with a purity of 95 ± 1% by area percent via liquid chromatography (LCAP). The CLLE efficiency was calculated to be 88 ± 2% for the rich solution and the final two steps afforded a yield of 90 ± 2% of ciprofloxacin via LCAP. The overall production was 700 g/24 h, meeting the project’s goals.

### 3.4. Linezolid

Linezolid (**4**) is a synthetic oxazolidinone antimicrobial drug. It is indicated for use in Gram-positive infections and approved for the treatment of bacterial pneumonia, skin and skin-structure infections, and vancomycin-resistant enterococcal (VRE) infections. It represents the last line of defense against multi-drug-resistant Gram-positive bacteria. It was discovered and developed by Pfizer in the 1990s and approved by the FDA in April 2000. It is sold under the brand name Zyvox^TM^ and appears on the World Health Organization’s List of Essential Medicines [98,122,123,124,125,126,127,128].

Russell et al. have demonstrated an effective strategy for the synthesis of this drug in a completely continuous fashion [86]. This study represents the highest number of reactions performed in sequential flow without the purification of intermediates and workup (Figure 5). While the reported synthetic routes require several steps and rely on several organic chemistry manipulations, they were able to develop the synthesis so that it can be performed in only seven steps. They have slashed the total time from more than 60 h in batch mode to 27 min in flow mode. The E-factor calculated for this process was 25, whereas the average for chemical reactions is 3.57, and in the pharmaceutical industry, it ranges from 25 to 100. Linezolid (**4**) was obtained at a yield of 73%, which corresponds to a throughput of 816 mg h^−1^ [129,130,131,132]. This work represents the application of the concept to challenging reactions, as it highlights how dangerous and hazardous reagents and solvents can be used in a continuous process in complete safety.

In the first step, the authors promoted a Ritter-type reaction between (+)-epichlorohydrin, acetonitrile and BF_3_∙OBu_2_ to obtain the nitrilium intermediate with a yield of 90%. 2-propanol was used to quench the Lewis acid in order to avoid clogging in the tubes and to generate the imidate intermediate. The epoxide intermediate was formed by treating the previous stream with lithium *tert*-butoxide in a 1:1 THF:1,2-dichloroethane (DCE) mixture. Overall, the residence time for these steps was 10.2 min. At this point, the authors focused on the nucleophilic aromatic substitution (S_N_Ar) between morpholine and 3,4-difluoronitrobenzene and on hydrogenation using a mass-flow controller (MFC) to obtain 3-fluoro-4-morpholinoaniline. They used the 1,4-dioxane and *N,N*-dimethylformamide solvent system to enhance the S_N_Ar rate, to solubilize all of the starting materials and byproducts and ensure good compatibility with the palladium packed bed. The hydrogenation reactor was a compact stainless steel packed bed of Pd(0) that was operated at 100 °C and 100 psi of back-pressure. After hydrogenation, the hydrogen-gas excess was removed through the continuous surge separator, and the aniline stream was introduced into the epoxide stream, giving the last intermediate without any additional activating agent. In the final step, linezolid (**4**) was generated by treating the stream with *N,N*-carbonyldiimidazole (CDI) and subsequent offline crystallization.

### 3.5. Tazobactam

Tazobactam (**5**) is a β-lactamase inhibitor used in combination with β-lactam antibiotics. It was first marketed in the USA in 1992 with piperacillin under the brand name Tazocin^TM^. In recent years, tazobactam (**5**) has been studied in combination with other β-lactam drugs due to its low toxicity and strong activity in fighting antimicrobial resistance (AMR). It appears on the World Health Organization’s List of Essential Medicines [98,133,134,135,136].

Zhou and coworkers have reported an interesting tazobactam (**5**) synthesis that works as a combination of continuous-flow and batch conditions [87]. Flow chemistry was implemented in the first three steps and in the final step, giving a total yield of 37.1%, whereas the yield was 30.1% in batch mode (Figure 6). This synthetic route was safer and more efficient, as it provided a 7% reduction in process mass intensity (PMI) while maintaining high purity (99.8%).

The 6-monobromopenicillanic acid intermediate was obtained in a two-phase system using dichloromethane as the solvent, thus avoiding potassium bromide precipitation. The yield reached 88.2%, whereas it was 90% in batch mode. In order to ensure a high yield in the protection step with diphenylmethanol, the authors investigated the correct order in which to add the reagents 1-(3-dimethylaminopropyl)-3-ethylcarbodiimide hydrochloride (EDC) and 4-dimethylaminopyridine (DMAP). As reported in the Scheme, EDC was added to a rich solution of the 6-monobromopenicillanic acid intermediate, while other reagents were added to a T-shaped mixer. In this way, they obtained a yield of 92.4% of the 6-monobromopenicillanic acid intermediate in benzhydryl ester. The reaction with peroxyacetic acid to furnish the sulfoxide was carried out at room temperature, demonstrating the safety and energy savings of flow chemistry. The authors also implemented a flow process to produce, in situ, the peroxyacetic acid used in this system. Finally, the flow reactor was applied in the final step to deprotect the carboxylic group in the presence of *m*-cresol, which was used as solvent and scavenger, giving the aforementioned overall yield.

### 3.6. Key Vaborbactam Intermediate

Vaborbactam is part of the new generation of β-lactamase inhibitors, classified as a non-β-lactam β-lactamase inhibitor. It is a cyclic boronic acid that was discovered, in 2015, by Rempex Pharmaceuticals, which is a ‘The Medicines Company’ subsidiary and now part of Melinta Therapeutics. It is combined with meropenem for the treatment of complicated urinary-tract infections and pyelonephritis. It was approved by the FDA on 29 August 2017 and sold under the brand name Vabomere^TM^. It acts against serine carbapenemase enzymes, including *Klebsiella pneumoniae carbapenemase* (KPC), boosting carbapenem action. This drug is administered via intravenous injection into a vein and appears on the World Health Organization’s List of Essential Medicines [98,137,138,139,140,141,142,143].

Stueckler et al. have presented a flow approach for the synthesis of the key intermediate (**6**) of this inhibitor [88]. They moved the Matteson reaction from batch mode to flow mode while improving diastereoselectivity, purity, reproducibility, yield and productivity for the key intermediate (**6**). This flow system is currently applied in the industrial-scale production of the intermediate under cGMP conditions, with several hundred kilograms being manufactured, and has been approved by FDA inspection.

In batch mode, the need to cool the process to −95 °C, the need to remove reaction heat, the high dilution and slow dosing protocols were impediments to commercial production. Moreover, byproduct formation, due to poor mixing, limited productivity. The authors overcame these issues using the patented flow technology depicted in Figure 7 [144].

In the first heat exchanger, dichloromethane was introduced with THF and cooled to −78 °C. Similarly, *n*-BuLi in heptane was mixed with THF and cooled to −78 °C. The use of THF as a cosolvent was necessary to prevent the precipitation of *n*-BuLi at low temperature. The (dichloromethane)lithium that was formed in the first flow reactor reacted with the substrate in heptane and zinc chloride, used as a Lewis acid, to obtain the intermediate. The two Matteson homologation steps are borate-complex formation and rearrangement with the concomitant stereoselective displacement of one chlorine. The customized continuous loop quench application allowed the process to be operated at a higher temperature (T ≥ −10 °C) than in the previous conformation (T < −20 °C), facilitating industrial production with improved economics and reduced ecological footprint. With this technology, Thermo Fisher Scientific can synthesize this intermediate (**6**) at a yield of more than 97%, a chemical purity of more than 98% area by HPLC and a diastereomer ratio of above 95:5.

### 3.7. Key Cefodizime Intermediate

Cefodizime is a third-generation cephalosporin with broad-spectrum activity against aerobic Gram-positive and Gram-negative bacteria. It is administered intravenously and intramuscularly to treat upper- and lower-respiratory-tract infections and urinary-tract infections. A single dose contains 1 g or 2 g of the drug, which is used for an average of 7-to-10 days. It is not currently approved by the FDA for use in the USA [145,146,147].

Wirth et al. have published an interesting manuscript for the continuous synthesis of thiazolyl-7-aminocephalosporanic acid (7-TACA, **7**) [89]; the key cefodizime intermediate. This involves a 3′-modification using methylmercaptothiazolyl acid (MMTA) on 7-ACA, the antibiotic’s backbone, as depicted in Figure 8.

In this system, the reagents were dissolved in phosphate buffer solution at pH = 7.0 and room temperature. The 7-ACA concentration was 100 mmol/L, while 1.0 equivalent of MMTA was used. The peristaltic pumps (P100 and P200) transported the raw-material solution to the Y-shaped mixer (M100) in flow towards the reactor. The flow reactor had an inner diameter of 3.1 mm and was submerged in an oil bath, which could be heated to 180 °C. The output was collected in a vessel in order to analyze the product in solution. The common batch method runs at 60 °C for one hour. These reaction conditions led to a colored compound. Chromatographic treatment is therefore essential to achieving the requested specifications. By taking advantage of the high heat-transfer capacity and small dimensions of the system, the authors worked at higher reaction temperatures and very short times. This concept is fundamental to the application of flow chemistry in cephalosporin synthesis because of the instability of the cephalosporin nucleus and its degradation at high temperature. The authors studied the parameters using DOE and obtained a yield of 85% and a selectivity of 85.3% for 7-TACA (**7**), working at 119 °C with a residence time of 3.99 min at pH = 7.0–7.5. In the continuous process, the thermal stress was lower, and the results were comparable to batch production. The space-time yield increased by a factor of 130.

## 4. Antibiotic Building Blocks in Flow Mode

In this section, we report the synthesis of the specific building blocks needed in antibiotics production. Improving safety, scalability and process intensification are the key drivers for the use of this technology. These concepts are attractive for industrial chemists that are faced with deciding between process development in batch or flow mode.

### 4.1. β-Methyl Vinyl Phosphate (MAP) as a Building Block for β-Methyl Carbapenem

Carbapenems play a critically important role in our antibiotic armamentarium. Of the many hundreds of different β-lactams, they possess the broadest activity spectrum and greatest potency against Gram-positive and Gram-negative bacteria. They are often used as “last-line agents” or “antibiotics of last resort” when patients become gravely ill or are suspected of harboring resistant bacteria [80,148]. β-Methyl vinyl phosphate (MAP, **8**) is an advanced intermediate for the β-methyl carbapenem class, which includes meropenem, ertapenem, doripenem and tebipenem. In the batch process, the rhodium-catalyzed insertion of a carbene generates the carbapenem fused-ring system. Subsequent reaction with diphenyl chlorophosphate (DPCP) in the presence of a base (*N,N*-diisopropylethylamine, DIPEA) and catalytic DMAP affords MAP (**8**) in 85–87% yields after crystallization. The drawbacks to commercial production include the difficulty in recovering the valuable rhodium and solvents. As the first step is performed with Rh_2_(Oct)_4_ in a homogeneous environment, recovery is typically 70%, meaning that organic-solvent incineration is required.

To overcome the shortcomings, Gage et al. have published a flow process for carbenoid N-H insertion and phosphorylation that is capable of producing 100 kg of MAP (**8**) per batch in cGMP manufacturing (Figure 9) [90].

They initially studied several methods to immobilize the Rh(II) complex on organic polymers for use in a packed-bed reactor. A polymeric support generated from *p*-vinylbenzyl alcohol and sebacic acid was chosen for its excellent swelling characteristics and good mechanical stability. This support facilitated higher metal recovery (>90% versus ≈70% for batch mode) and complete solvent recovery. In addition, it was possible to use the catalyst six times without any appreciable activity loss, with Rh leaching dropping to 0.99% per run. Given that Rh-catalyzed cyclization is a gas–liquid–solid three-phase reaction, as nitrogen gas is released, the authors investigated all of the parameters in order to achieve good quality and yields. They obtained a yield of 97% for the cyclization intermediate, working at residence times in the 7–12 min range and at 45 °C. They also determined that a reactor length of 320–600 mm was capable of working at 0.15 MPa. ZnBr_2_ was employed as the stabilizer at the onset of cyclization. Even the phosphorylation was studied, and an in-solution yield of 95% for MAP (**8**) was achieved with the cyclization substrate, DPCP, DMAP and DIPEA all well-mixed in the process, and with the same temperature as the batch process being used. This work culminated in three validation batches that produced an isolated yield of 80%, which was slightly lower than the optimized batch yield.

### 4.2. 6-Aminopenicillanic Acid (6-APA)

6-Aminopenicillanic acid is an important precursor for modern β-lactam antibiotic synthesis. It was discovered by scientists at Beecham, now GlaxoSmithKline (GSK), in 1958 via the fermentation of penicillin. It is currently produced in the enzymatic reaction of penicillin G (PenG), using penicillin acylase (PA) as the enzyme [149,150].

Vobeckà and coworkers have employed an aqueous two-phase system (ATPS) with an integrated microfluidic platform to synthesize 6-APA, as reported in Figure 10 [91].

In this system, the ATPS was made up of 15 wt% of PEG 4000, 10 wt% of phosphate buffer, to reach pH = 8.0, and 75 wt% of water, and separated 6-APA from the byproduct phenylacetic acid (PAA) and the enzyme (PA). While PenG hydrolysis occurred in the microchannels, the extract phase was inputted in counter-current flow to separate the product through a membrane. The correct molecular weight cut-off provided the two flows with hydrodynamic stabilization and a direct current electric field imposed perpendicularly to the fluids intensified 6-APA and PAA transport to the top phase. The enzyme-containing bottom phase could be recycled continuously to minimize the time and enhance the product yield. As indicated by the authors, this work needs to be optimized to reach high yield and productivity, but represents a starting point for adapting the flow chemistry concepts to a building block discovered more than fifty years ago.

## 5. Conclusions

This brief overview has presented the cases in which flow chemistry has been used in the synthesis of antibiotics. Nowadays, the technique has not only matured as an enabling technology, but is accepted and advocated by scientists as a technology to take the synthesis of active ingredients or natural products to a higher level. The advantages of miniaturization, leading to improved heat and mass transfer, are generally used to achieve better process control and selectivity in various chemical reactions. Some processes have been studied at the laboratory scale, while others have been scaled-up to industrial scale and validated by the FDA. It is expected that pharmaceutical companies will continue to engage in this area to innovate antibiotic synthesis while protecting the environment and workers.

## Data Availability

Not applicable.

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
