# Peer review of "Towards Antibiotic Synthesis in Continuous-Flow Processes"

_molecules, 2023, doi:10.3390/molecules28031421_

Round 1

Reviewer 1 Report

I enthusiastically support the publication of this article, which is well-written, timely, and reports innovative science. I appreciated the introductory overview of the topic, the thorough bibliography reported, and the connections between green chemistry and green engineering in the synthesis of small molecules.

I suggest standardizing the size of the molecules and the size of the characters related to atoms in Table 1. Some typos should be corrected (spaces between number and unit in rows 125, 126, 149, etc. are missed).

Schemes 3, 4, and 5 should be unified in the size of molecules compared to the previous ones. Scheme 5 is of low quality. 

Author Response

Authors warmly acknowledge reviewers and Editor. The manuscript was thoroughly revised by a native English speaker.

Reviewer 2 Report

Dear Dr,

the introduction is good.
please check plagiarism.
 the English level is suitable.
in Methods :please, The literature search show with a chart.
Please specify the time range of review.

Author Response

Reviewer and Editor are kindly acknowledged
